# The Impact of Using Kinesio Tape on Non-Infectious Complications after Impacted Mandibular Third Molar Surgery

**DOI:** 10.3390/ijerph18020399

**Published:** 2021-01-06

**Authors:** Aleksandra Jaroń, Olga Preuss, Elżbieta Grzywacz, Grzegorz Trybek

**Affiliations:** Department of Oral Surgery, Pomeranian Medical University in Szczecin, Powstańców Wielkopolskich 72/18, 70-111 Szczecin, Poland; olga.preuss@pum.edu.pl (O.P.); 50422@student.pum.edu.pl (E.G.); grzegorz.trybek@pum.edu.pl (G.T.)

**Keywords:** third molar removal, kinesio taping, pain, swelling, edema, trismus, morbidity, complications

## Abstract

Non-infectious complications such as post-extraction pain, trismus, and swelling are extremely common after impacted wisdom tooth removal. The aim of the study was to assess the impact of using kinesio tape on the level of the postoperative swelling of soft tissues, trismus, and pain in patients undergoing the surgical extraction of an impacted mandibular third molar. One hundred patients undergoing the surgical extraction of a lower wisdom tooth were randomly divided into two groups: a study group with kinesio taping (KT) (*n* = 50) and a control group without kinesio taping (NKT) (*n* = 50). The surgical procedure was performed according to the same repeatable scheme. Kinesio tape was applied immediately after surgery in the KT group. In both groups, measurements of swelling, trismus, and pain were performed before the surgery and on the third and seventh postprocedural days. Kinesio tape had a significant effect on the decrease in facial swelling on the third day after surgery and a decrease in trismus and pain severity levels on the third and seventh days after surgery. The kinesio tape method is non-invasive, continuously active throughout the entire application period, and requires no additional patient appointments. KT application is an effective method for reducing postoperative edema, pain, and trismus after impacted mandibular wisdom teeth surgery.

## 1. Introduction

The removal of an impacted third molar in the mandible is the most common surgical procedure performed in the field of oral surgery [1,2,3,4]. The decision to perform surgery to remove an impacted lower wisdom tooth should always be carefully considered, and all contraindications and possible postoperative complications should be analyzed [5]. The surgical removal of an impacted third molar is one of the most complicated procedures in the field of oral surgery [5,6]. The degree of difficulty depends, inter alia, on the degree of tooth impaction, the tooth’s anatomical structure, and the tooth’s location in relation to adjacent structures [7]. The procedure is associated with the risk of infectious and non-infectious postoperative complications.

The triad of the most common non-infectious complications, i.e., post-extraction pain, trismus, and the swelling of soft tissues, results from surgical trauma and related procedures traumatizing periodontal tissues. Traumatizing procedures include: the incision of the mucosa, the detachment of the mucoperiosteal flap, the removal of the bone casing, and the retraction of the surgical field by the surgical hook, which usually causes the additional local stagnation of the lymph [3]. There are few methods of preventing these complications. The effectiveness of some of them is questionable, while others might exert side effects. Therefore, there is currently no effective and safe method that can significantly lower the risk of these complications. The latest reports have indicated that the use of kinesio tape could have a significant effect on reducing pain, swelling, and post-operative trismus. The results of previous observations in this area are promising; however, the number of available studies is very small, as is the size of the patient groups discussed in them [8].

A physical method of treating selected postoperative complications using so-called dynamic taping (kinesio taping—KT) has been recently postulated in oral and maxillofacial surgery. It is a therapeutic method that is widely used in sports medicine and the rehabilitation of the musculoskeletal system. It is used to prevent injuries, reduce pain after an injury, and increase the range of motion in joints, as well as to functionally increase performance [9,10].

Kinesio tapes are thin, flexible tapes that can be stretched by up to 30–40% of their original length. After their application to the skin, they do not limit mobility of the body area receiving treatment. They are made of cotton covered with hypoallergenic glue. The thickness, specific weight, and extensibility of the tapes are similar to the properties of the epidermis [11,12,13]. The mechanism of the tape is based on supporting the body’s natural self-healing processes [14]. The therapeutic effect of the tape is to improve the flow of blood and lymph at the site of its application by pulling the skin away from the muscles and subcutaneous tissue, as well as by causing pressure and stretching the skin, which lead to the activation of mechanoreceptors through the central nervous system, which leads to increased excitability muscles [15]. A decrease in the level of perceived pain occurs due to the reduction of pressure on nociceptors [16]. The tapes are waterproof, air-permeable, and do not disturb thermoregulation. They should be kept on the skin for four-to-five days [17]. Kinesiology taping is a non-invasive and cheap method. Another advantage is its continuity of therapy, which allows it to exert a 24 h therapeutic effect.

In the literature, there have been few reports on the effectiveness of kinesio taping after surgical tooth extraction [8]. The positive results of preliminary reports on the use of kinesio tape in the surgery of the lower wisdom tooth became an inspiration to research our clinical material. The aim of the study was to assess the impact of kinesio taping on the level of postoperative swelling of soft tissues, trismus, and pain in patients undergoing a surgical extraction of an impacted mandibular third molar.

## 2. Materials and Methods

The research was carried out at the Department of Oral Surgery after obtaining a positive opinion from the Bioethics Committee (opinion no. KB-0012/152/13, as amended. KB-0012/135/15)

### 2.1. Patient Recruitment and Study Groups

Participation in the research project was offered to 100 consecutive Caucasian patients (male—26; female—74). The study group included 36 women and 14 men, while the control group included 38 women and 12 men undergoing the surgical removal of an asymptomatic, impacted mandibular third molar, usually for orthodontic reasons. The exclusion criteria included: age below 18 years; unstable arterial hypertension; pregnancy; allergy to local anesthetics; patients with contraindications for outpatient surgery; pericoronitis; and those who, based on a preoperative radiological examination, needed no bones removed during surgery.

The eligible patients were randomly assigned to one of the two groups:
Study group (*n* = 50): kinesio tape applied for five days after surgeryControl group (*n* = 50): non-kinesio tape applied.

Patients in each group were asked about their age and gender.

### 2.2. Surgery

The surgical procedure was performed according to the same repeatable scheme for each patient by one of the two oral surgery specialists with the same degree of professional experience. The surgical procedure was performed under the local anesthesia of the inferior alveolar nerve and the buccal nerve using 2% lidocaine with 1:80,000 noradrenaline. The incision was made on the ridge of the mandibular alveolus lateral from the linea obliqua towards the second molar. Then, a release incision was made in the distal part of second molar, downward and in mesial direction for approximately 1 cm according to the same repeatable scheme for each patient. The bone casing of the mandible was removed using a drill mounted on the handpiece with cooling including a sterile physiological saline solution according to the same protocol for each patient. After the surgery, the wound was closed with sutures for a period of 7 days. A risk of increased bleeding was not expected. No additional hemostatic materials were used to exclude and minimize the influence of other factors on the test results. The time of each wisdom tooth removal was measured in minutes. In addition to the standard recommendations for the procedure after the tooth extraction, patients in both groups were treated with an antiseptic rinse based on 0.1% chlorhexidine solution, as well as ketoprofen in a 100 mg dose taken twice daily, were on a semi-liquid diet, and avoid physical exertion for seven days after treatment.

### 2.3. Kinesiotaping

Kinesio Tape K-Active Tape Classic (Nitto Denko Corporation, Osaka, Japan; K-Active Europe GmbH, Hösbach, Germany), 50 mm × 5 m was applied immediately after the wisdom tooth removal in the KT group. The skin before application was cleaned with sterile gas soaked in saline. For every particular patient, the tape was prepared before its application. In order to apply the tapes, a technique mapped from the technique used by Ristow et al. was employed [18]. The 50 mm-wide tape was cut into 3 equal pieces forming smaller stripes. The application of the tape was started in the area of supraclavicular lymph nodes. The tape was then advanced to line A on the patient’s face where the greatest edema was expected. The method of application is shown in Figure 1. The degree of tension in the tape was approximately 15% of the maximum stretch. Tape length (x) was calculated by Equation (1). The distance from the area of supraclavicular lymph nodes to line A on the patient’s face (y) was 115%. The tape length needed to create 15% tension was calculated from the formula:(1)$$\frac{100\%\;\times \;y\;\left[\mathrm{cm}\right]}{115\%}.$$

The tape was prepared according to the calculations, which was about 86.96% of the y-distance. During application, the tape was stretched in such a way that it spanned the entire y-distance. The tapes were kept for five days after surgery. In both groups, measurements of swelling, trismus, and pain were performed before the surgery and on the 3rd and 7th postprocedural days.

### 2.4. Data Collection

#### 2.4.1. Edema

For detailed evaluations of the post-operative edema of the facial soft tissues, appropriate measurements of the patient’s face were taken before each procedure (Figure 2). Each measurement was performed with an elastic measuring tape. All patients were located in the same position. The measurements of swelling were performed using five lines mapped out on the patient’s face. All points were connected with the lines.
Line A—from tragus (T) to cheilon (Ch)Line B—from tragus (T) to pogonion (WPg)Line C—from tragus (T) to exocanthion (Ex)Line D—from exocanthion (Ex) to gonion (Go)Line E—from gonion (Go) to nasion (N)

#### 2.4.2. Trismus

The maximum jaw opening—the distance between the incisal edges of the upper medial incisors and the incisal edges of the lower medial incisors in the midline of the body—was measured with a caliper.

#### 2.4.3. Pain

To assess the impact of the method on the postoperative pain level, each participant determined the pain level according to the VAS (Visual Analogue Scale). The patient marked a point which, in their opinion, corresponded to their pain level. The patient was asked to mark a point on a 100 mm horizontal line, with the left pole showing no symptoms at all and the right pole showing “unbearable” symptoms.

### 2.5. Statistical Methodology

Statistical analysis was performed with the R statistical package (The R Foundation for Statistical Computing, Indianapolis, IN, USA, 2012). The variables were described using position measures, i.e., arithmetic mean and quartiles with the median, standard deviation (SD), and the minimum and maximum values. In all analyses, the level of statistical significance was assumed to be 0.05. The qualitative variables were compared in both groups using the chi-quadrant test. A comparative analysis of edema between the two groups was performed using the Mann–Whitney test after the prior confirmation of the non-normality of the data distribution using the Shapiro–Wilk test. The differences between the groups with normal distributions of the variable were analyzed using Student’s *t*-test. On the other hand, the Mann–Whitney test was used if the values of the maximum jaw opening and pain symptoms did not show a normal distribution in the analyzed groups.

## 3. Results

### 3.1. Baseline Characteristics

The study group included a total of 100 patients: 50 patients with KT and 50 patients without KT. The study group included 36 women and 14 men, while the control group included 38 women and 12 men. The chi-square test did not show any significant differences in this respect between the two groups (*p* = 0.82). The age of the patients in the study group fluctuated in the range of 19–59 years (median of 26.5 years), while in the control group, it was 18–38 years (median of 25 years). Both groups showed no statistically significant differences for age (*p* = 0.221; Mann–Whitney test). The duration of the treatment in both groups was similar; in the study group, it ranged from 10 to 60 min, whereas in the control group, it ranged from 6 to 60 min. A statistical analysis showed no statistically significant differences in this respect between the two groups (*p* = 0.801; Mann–Whitney test). In the study group, a total of thirty teeth—38 and twenty teeth—48 were removed, while in the control group, it was 28 and 22, respectively. The chi-square test did not show any statistically significant differences between the two groups in this respect (*p* = 0.839). The complete characteristics of the sample with the results of statistical calculations are presented in Table 1.

### 3.2. Analysis of Postoperative Edema Measurements

The conducted analysis showed that on the third day after surgery, there was a statistically significant difference in the length of the D and E lines between both groups. The mean length of the D line in the control group was 10.35 cm, and for the E line it was 15.44 cm, whereas in the study group the mean lengths were 9.99 and 14.89 cm, respectively. The significance levels were lower than 0.05 and were *p* = 0.008 and *p* = 0.005 for the D and E lines, respectively. The analysis of the differences in the measurements of the lines from A to C showed no statistical significance. The detailed results of the statistical analysis in this regard are summarized in Table 2.

It was shown that the E line was significantly longer in the control group (*p* = 0.029; Mann–Whitney test). Both the median and the arithmetic mean were higher compared to the study group. Concerning the remaining lines (A–D), no statistically significant differences were found between the two groups on seventh day. The detailed results of the statistical analysis in this regard are summarized in Table 3.

### 3.3. Analysis of Postoperative Trismus Measurements

The measurement of the baseline size (before the procedure) of the jaw opening did not show any statistically significant differences between the two groups. The mean value in the study group was 4.73 cm (±0.72), whereas in the control group, it was 4.5 cm (±0.56). It was shown that both on the third and seventh days after the procedure, the jaw opening level was significantly higher in the study group. The mean jaw opening value on the third day after surgery among patients with tapes applied was 2.93 cm (±1.08), whereas in the group without tapes, it was 2.42 cm (±0.69). On the seventh day after surgery, the degree of trismus disappeared in both groups. It was reduced, but, in the study group, the average degree of jaw opening was still significantly higher (3.97 cm ±1.03) than in the control group, where it was 3.55 cm (±0.81).

The detailed results of the statistical analysis in this regard are summarized in Table 4.

### 3.4. Analysis of Postoperative Pain Measurements

An analysis of the initial (before the procedure) pain intensity level measured with the VAS showed no statistically significant differences between the two groups (*p* = 0.065; Mann–Whitney test). The mean value in the study group was 5.2 (±10.74), whereas in the control group, it was 1.6 (±5.48). A statistically significant difference in the level of pain was demonstrated between the study groups on the third day after the procedure (*p* = 0.003; Mann–Whitney test). In the study group, the average pain level measured by the VAS was 37.6 (±25.36), whereas in the control group, it was 52 (±23.82). However, no statistically significant differences concerning the discussed parameter were found between the patients in both groups on the seventh day after surgery. Table 5 presents the detailed results of the statistical analysis in this regard.

## 4. Discussion

Nowadays, the surgical extraction of a lower third molar is the most commonly performed oral surgery procedure. Due to phylogenetic changes in the jawbones, the incidence of retention of third molars is constantly increasing. This has contributed to an increase in the medical needs of the surgical extraction of wisdom teeth [1,2,3]. This procedure very often leads to postsurgical, non-infectious complications, such as facial swelling, trismus, and post-extraction pain.

While the risk of infectious complications in immunocompetent patients is low, the occurrence of certain non-infectious complications such as post-extraction pain, trismus, and the swelling of the soft facial tissues is extremely common in the postoperative period. Infectious complications in immunocompetent patients develop in less than 4% of cases and are mainly related to infections of the surgical wound [19]. They may appear in the form of an abscess [20] or osteitis, and they are much less frequently systemic complications [21]; therefore, they usually affect immunocompetent patients. Incidentally, there are long-term infectious complications associated with intraoperative bacteremia, but this applies to a small group of patients predisposed to, for example, infective endocarditis [22] or an infection of the implanted artificial joint [23].

Though non-infectious complications most often do not pose a threat to a patient’s health and are very common, they constitute a major therapeutic problem and significantly reduce the patient’s postoperative quality of life [1]. They are much more common in the extraction of lower wisdom teeth and are normally post-extraction pains, as well as trismus and the swelling of the surrounding soft tissues (edema). Among the much less common complications from the group of non-infectious agents, one should also mention post-extraction bleeding (sanguinatio post extractionem) and so-called dry alveolitis (alveolitis sicca) [1,2,3].

The methods currently used for these indications often lead to side effects associated with discomfort for the patient, especially in the case of pharmacotherapy (antibiotic, non-opioid analgesic, or steroid) [5]. Post-extraction pain is often an indication for taking analgesics, especially from the group of non-steroidal anti-inflammatory drugs (NSAIDs), which, in turn, are one of the most toxic groups of drugs to many systems and organs, especially the gastrointestinal tract [24]. The matter is further complicated by the fact that these drugs are often used in so-called self-medication (self-treatment carried out without medical supervision), which results in their abuse, an exceeding of maximum doses, and a risk of dangerous interactions with other medications taken by patients. Trismus, on the other hand, usually impedes speech and food intake, further reducing the postoperative quality of life in patients [25]. The location of edema in the visible area of the face means that after the surgical extraction of an impacted lower wisdom tooth, patients are often on sick leave and avoid contact with the environment. All this results in the fact that patients feel a great fear of possible treatments of this type.

There is a great need to introduce a minimally invasive method to prevent the occurrence of non-infectious complications after surgical wisdom tooth extraction. One of them may be the dynamic taping method (kinesiology taping), which, until recently, was only used in sports medicine [10,26,27]. Currently, it is assumed that the above-mentioned non-infectious complications constitute a physiological response of the body to surgical trauma [3]. According to Bortoluzzi et al., more than 50% of patients after the surgical extraction of an impacted lower wisdom tooth on the first days after surgery experience moderate-severe to severe pain [28]. Trismus of varying severity is observed in the majority of patients after surgery, and in an analysis of Osunde and Saheeb, it occurred in all patients under study [24].

Initially, KT was used mainly in physiotherapy practice and sports. Its action was mainly used to heal injuries. Currently, a small amount of research in the literature deals with KT in the maxillofacial area, especially with the third molars in the mandible [8].

Ristow [18] was the first author to introduce dynamic taping methods to the surgery of the mandibular third molar.

Authors have provided different taping techniques in their research centers, and some of them have not provided their technique [29]. Some authors have not provided the degree of tape tension. They applied tapes by pulling the skin away from the subcutaneous tissue, thus facilitating lymph flow (anti-edema effect) and reducing pressure on nociceptors (analgesic effect) [8].

Similarly to the report by Ristow et al. [18], in our study, the method of applying three tapes with a width of about 1.5 cm was used. In studies by Genc and Erdil, their tension was reduced to 15%, which is currently used in the lymphatic technique [14,30,31]. All the techniques mentioned by the authors contributed to the reduction of postoperative edema. It would seem reasonable to carry out studies comparing different techniques of KT application.

Researchers have disagreed about the length of time that patient should be taped. Genc and Erdil removed tapes after two days [32,33]; Tatli [34], Gözlüklü [35], de Rocha Heras [36], and Ristow [18] removed tapes after five days; and Yurttutan [37] removed tapes after seven days.

In this study, as in most publications, the tapes were removed after five days. Further research on the effectiveness of kinesio taping in counteracting post-ascending complications should be conducted because the literature in regard to the study groups (13–76 patients) is limited in number. In the original study, as many as 100 patients were analyzed. 50 of them were in the study group and 50 in the control group. In the literature, the authors have unanimously agreed that the most severe swelling occurs after two-to-three days following the surgical removal of the third molar in the mandible [38]. Most authors also agree on the seven-day follow up [18,29,32,33,34,35,37,39,40].

In our study, the removal of the third molar was performed unilaterally. A study of the application of two therapies or a comparison of the application of KT therapy to a procedure without the use of KT in one patient in split-mouth studies could give the best results due to the possibility of comparing the therapeutic effect. Each patient feels pain differently, which may cause inaccuracies in results. In our study, as in most of the studies presented in the literature, the pain sensation was determined with the VAS to standardize this subjective feeling.

Ristow showed significantly less pain, trismus, and swelling after the surgical extraction of the wisdom tooth on the third day after surgery in the group with KT complications than in the control group [18]. Similar conclusions in the split-mouth randomized controlled trial (RCT )study were reached by da Rocha Heras et al. [36]. The authors also used other methods of applying the tapes. Gozuklu et al. used a modified method of KT application, the effectiveness of which they compared with the classic application [35]. The new method was more effective in reducing postoperative complications than the classic method [35].

The cited data show that the kinesiology taping method significantly reduces the level of soft tissue swelling after maxillofacial surgery, and it is also likely to reduce postoperative pain and trismus [8].

Some of the articles concerned the influence of patients’ age and the duration of the surgical removal of wisdom teeth on the severity of oedema, jaw compression, and pain after surgery [40]. A study by Bello et al. did not show any significant statistical effect of the duration of surgery on postoperative swelling and maxillofacial pain (*p* > 0.05). It was also revealed that the level of postoperative pain was statistically significant, depending on the duration of the procedure, as a longer time was associated with more severe pain [41].

In our study, due to the fact that the study and control groups did not differ in terms of age and duration, it can be concluded that only the use of kinesio tape patches had an effect on parameters such as maxilla, pain, and swelling.

Though the study group was not very large (*n* = 100), to the authors’ knowledge, it was the largest of the reported studies so far. In addition, the study could be expanded to include the determination of the degree of retention of the third molar in the mandible, as well as the classification of the position of the impacted third molar in relation to inferior alveolar nerve (IAN).

It should also be taken into account that few authors have reported the results of edema measurement with and without KT as a mean of all linear measurements [32,33,35,36,37,39,40]. In our study, as well as in Tatli’s study [34], the results were related to each linear measurement separately, which allowed us to separately determine the effect of KT on the occurrence and size of edema in each area. The effect of KT on the occurrence of postoperative pain was also emphasized: KT applications reduced the pressure on nociceptors (analgesic effect). Because flexible KT bands are not invisible, the placebo effect in the assessment of post-treatment pain cannot be ruled out. Patients’ lack of credibility was excluded in the Tatli study on a group of 60 patients. The authors in the study group used the placebo effect to rule out patients’ unreliability [34]. In our study, the edema was measured with an elastic band, but establishing precise measurement points increased the reliability of the results. Introducing volumetric measurements using 3D scanners and by comparing the color maps of stereolithography (STL) files obtained during scanning would allow one to achieve an even greater minimization of the risk of measurement errors [42,43,44].

## 5. Conclusions

The kinesio tape method is non-invasive, continuously active throughout the entire application period, and requires no additional patient appointments. Kinesio tape application after the surgical extraction of the third lower molar has a significant effect on decreasing facial swelling on the third day after surgery, decreasing the trismus level on the third and seventh days after surgery and decreasing pain severity on the third and seventh days after surgery. The KT application is an effective method for reducing postoperative edema, pain, and trismus after impacted mandibular wisdom teeth surgery.

## Figures and Tables

**Figure 1 ijerph-18-00399-f001:**
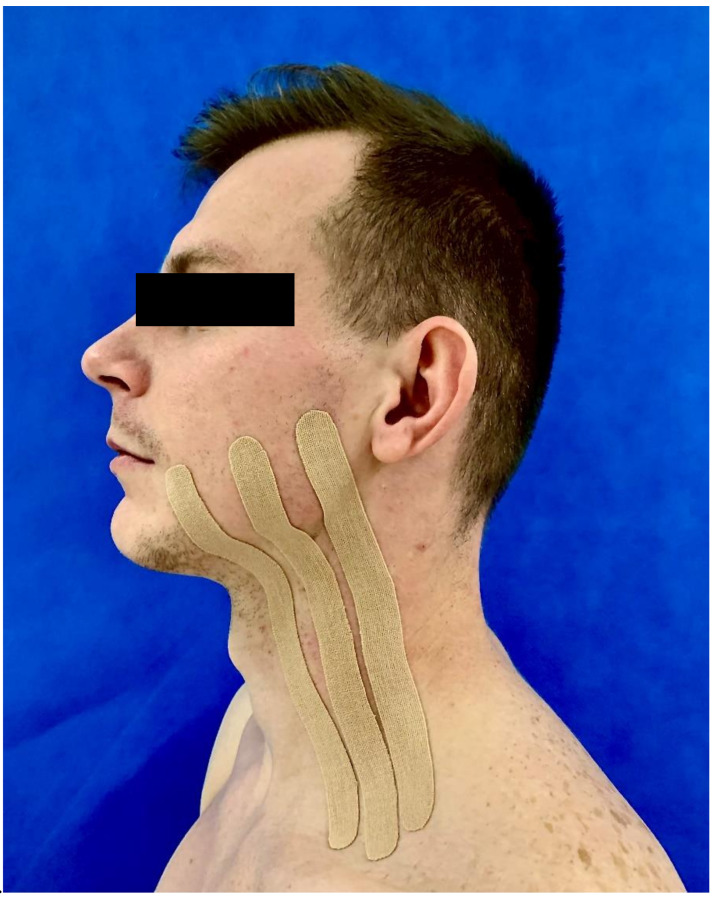
Method of kinesio tape application.

**Figure 2 ijerph-18-00399-f002:**
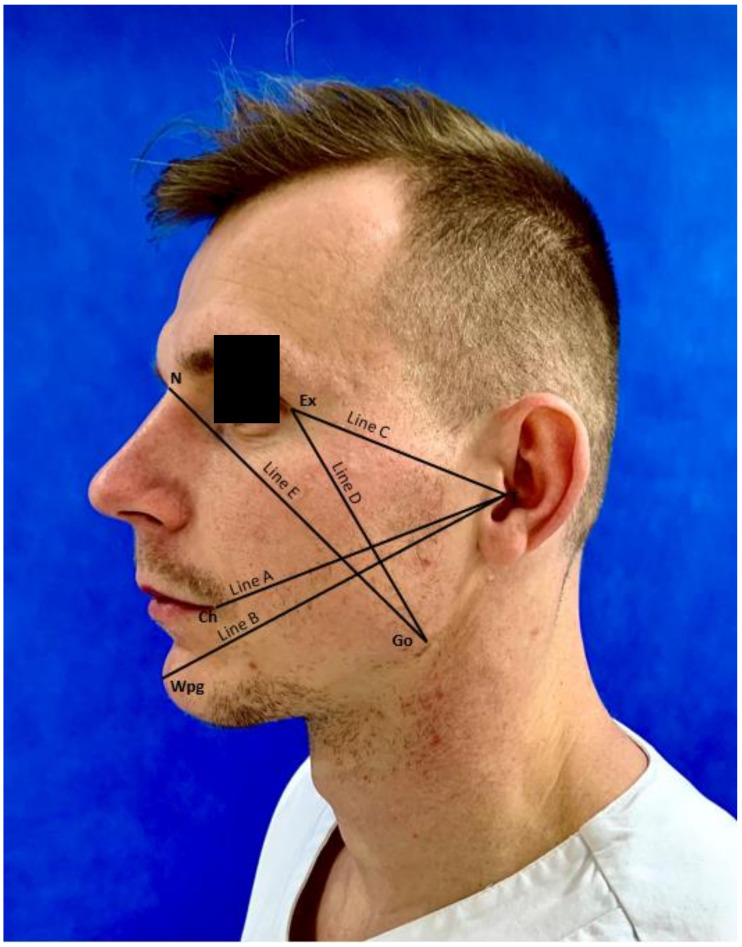
Measurement points and lines.

**Table 1 ijerph-18-00399-t001:** Characteristics of the study group.

Feature		Study Group (*n* = 50)		Control Group (*n* = 50)		Total (*n* = 100)		*p* *
		Median	Min–Max	Median	Min–Max	Median	Min–Max	
Age		26.5	19–59	25	18–38	25.5	18–59	0.221
Operation time [min]		21	10–60	24.5	6–60	23	6–60	0.801
**Feature**		***n***	%	*n*	%	*n*	%	***p* ****
Sex	Woman	36	72.00%	38	76.00%	74	74.00%	0.82
	Man	14	28.00%	12	24.00%	26	26.00%	
Tooth number	38	30	60.00%	28	56.00%	58	58.00%	0.839
	48	20	40.00%	22	44.00%	42	42.00%	

* Not normal variable distribution, *p* value from the Mann–Whitney test; ** chi-square test; *n*—number of patients; Min—minimum value; Max—maximum value; and *p*—significance level.

**Table 2 ijerph-18-00399-t002:** Measurement results of A–E line on the 3rd day between the two groups.

Measurement on the 3rd Day (cm)	Group	*n*	Mean	SD	Median	Min	Max	Q1	Q3	*p* *
Line A	study	50	11.75	0.78	11.55	10.45	13.5	11.1	12.4	*p* = 0.336
	control	50	11.89	0.91	11.78	10	14.5	11.45	12.5	
Line B	study	50	15.42	1.07	15.07	13.6	18	14.56	16	*p* = 0.304
	control	50	15.52	0.84	15.43	14	18.05	15	16	
Line C	study	50	8.42	0.89	8.35	6.95	11.25	7.78	8.91	*p* = 0.724
	control	50	8.45	0.67	8.22	7	10	8	9	
Line D	study	50	9.99	0.75	10	8.5	12	9.5	10.45	*p* = 0.008
	control	50	10.35	0.79	10.22	8	12	10	10.97	
Line E	study	50	14.89	0.9	15	12.5	16.6	14.16	15.5	*p* = 0.005
	control	50	15.44	0.9	15.5	13.5	16.9	14.75	16.24	

* Mann–Whitney test; *n*—number of patients; SD—standard deviation; Min—minimum value; Max—maximum value; Q1—first quartile; Q3—third quartile; *p*—significance level.

**Table 3 ijerph-18-00399-t003:** Measurement results of line A–E on 7th day between the two groups.

Measurement on the 7th Day (cm)	Group	*n*	Mean	SD	Median	Min	Max	Q1	Q3	*p* *
Line A	study	50	11.52	0.76	11.5	10	13.4	11	12	*p* = 0.964
	control	50	11.53	0.87	11.5	9.75	13.75	10.96	12.04	
Line B	study	50	15.15	1.13	15	12	18	14.5	15.76	*p* = 0.714
	control	50	15.02	0.86	15	13.25	17.05	14.5	15.57	
Line C	study	50	8.2	0.74	8.15	7	10	7.55	8.66	*p* = 0.732
	control	50	8.22	0.54	8.07	7	9.25	8	8.5	
Line D	study	50	9.8	0.77	10	8.25	11.5	9.25	10.15	*p* = 0.227
	control	50	9.93	0.93	10	7.75	11.5	9.51	10.57	
Line E	study	50	14.57	0.95	14.53	12.1	16.6	14	15.24	*p* = 0.029
	control	50	15	0.94	15	12.8	16.5	14.26	16	

* Mann–Whitney test; *n*—number of patients; SD—standard deviation; Min—minimum value; Max—maximum value; Q1—first quartile; Q3—third quartile; *p*—significance level.

**Table 4 ijerph-18-00399-t004:** The results of comparative analysis of the jaw opening before the procedure on the 3rd and 7th days.

Jaw Opening (cm)	Group	*n*	Mean	SD	Median	Min	Max	Q1	Q3	*p* *
Baseline	study	50	4.73	0.72	4.62	3.5	6.5	4.01	5	*p* = 0.135
	control	50	4.5	0.56	4.43	3.75	6.1	4	4.93	
3rd day	study	50	2.93	1.08	3	1	5	2.02	3.73	*p* = 0.012
	control	50	2.42	0.69	2.5	1.25	4	1.85	2.79	
7th day	study	50	3.97	1.03	4.08	1.5	6.35	3.06	4.74	*p* = 0.02
	control	50	3.55	0.81	3.6	1.5	6	3	4	

* Mann–Whitney test; *n*—number of patients; SD—standard deviation; Min—minimum value; Max—maximum value; Q1—first quartile; Q3—third quartile; *p*—significance level.

**Table 5 ijerph-18-00399-t005:** The results of the comparative analysis of the pain intensity level measured with the Visual Analogue Scale (VAS) before the procedure, on the 3rd and 7th days after the procedure between the control group and the study group.

Level of Pain Intensity (VAS)	Group	*n*	Mean	SD	Median	Min	Max	Q1	Q3	*p* *
Baseline	study	50	5.2	10.74	0	0	30	0	0	*p* = 0.065
	control	50	1.6	5.48	0	0	20	0	0	
3rd day	study	50	37.6	25.36	30	0	90	20	50	*p* = 0.003
	control	50	52	23.82	50	0	100	40	70	
7th day	study	50	16.8	20.04	20	0	80	0	27.5	*p* = 0.06
	control	50	25	22.7	20	0	80	0	47.5	

* Mann–Whitney test; *n*—number of patients; SD—standard deviation; Min—minimum value; Max—maximum value; Q1—first quartile; Q3—third quartile; *p*—significance level.

## Data Availability

The data presented in this study are available on request from the corresponding author.

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
