# Peer review of "The Impact of Using Kinesio Tape on Non-Infectious Complications after Impacted Mandibular Third Molar Surgery"

_ijerph, 2021, doi:10.3390/ijerph18020399_

Round 1

Reviewer 1 Report

The authors of the manuscript try to describe the application of Kinesio Tape after third molar surgery in the mandible. The Introduction is too detailed in some points and has a lot of overlaps with the Discussion. The Kinesio-Taping historical background is unnecessary and it does not give more information about the method used in the article. There are some references which appear in an unaccepted way in the Introduction and Discussion as well. There are some questionable data in Material and Methods, like "the degree of tension in the tape was approximately 15% of the maximum stretch" This is a quite important data based on the discussion and it is not given precisely. No data can be found in the Material and Method chapter about the gender numbers, then in the Results, detailed analysis happen. Although the chi-square test is a known statistical method, it is not mentioned in the Material and Methods, but data can be found in Results with this method.  Authors describe a statistical differences in 3.2. Analysis of Postoperative Edema Measurements  in the length of D line, but in the same Chapter they state "remaining lines (A–D), no statistically significant differences were found". The significant differences which are calculated are not convincing considering the gender ration and the different face characteristics of the patients. The Discussion chapter is short a and it is not appropriately discuss the results. It is just a brief review of the field without showing the significance of these results. 

Author Response

Reviewer 1

Dear Reviewer,

We would like to thank you for your valuable comments on the article. Below you will find our reply to your review. All changes are with a description or a comment and changes have been made to the manuscript. Changes are marked with yellow.

The authors of the manuscript try to describe the application of Kinesio Tape after third molar surgery in the mandible. The Introduction is too detailed in some points and has a lot of overlaps with the Discussion. The Kinesio-Taping historical background is unnecessary and it does not give more information about the method used in the article.

Thank you for comment. The introduction has been shortened and some information has been moved to the discussion section. Due to the multitude of corrections in discussion section, they were not marked in the text.

There are some references which appear in an unaccepted way in the Introduction and Discussion as well.

Thank You for comment. References have been improved.

There are some questionable data in Material and Methods, like "the degree of tension in the tape was approximately 15% of the maximum stretch" This is a quite important data based on the discussion and it is not given precisely.

Tape length (x) was calculated by equation (1). The distance from the area of supraclavicular lymph nodes to line A on the patient's face (y) was 115%. The tape length needed to create 15% tension was calculated from the formula:

no. 1 on text (1).

The tape was prepared according to the calculations, which was about 86.96% of the y-distance. During application, the tape was stretched in such a way that it spanned the entire y-distance.

No data can be found in the Material and Method chapter about the gender numbers, then in the Results, detailed analysis happen.

Thank You for the remarks. I introduced your suggestion to the text. 

Although the chi-square test is a known statistical method, it is not mentioned in the Material and Methods, but data can be found in Results with this method. 

Thank You for the remarks. I introduced your suggestion to the text. 

Authors describe a statistical differences in 3.2. Analysis of Postoperative Edema Measurements  in the length of D line, but in the same Chapter they state "remaining lines (A–D), no statistically significant differences were found".

Thank You for the remark. Statistically significant differences did not occur on the 7th day after the procedure. I added this information.

The significant differences which are calculated are not convincing considering the gender ration and the different face characteristics of the patients.

All patients were Caucasian adults. No visible differences in facial structure were observed.The results were related to the whole study group, without reference to gender, there were no statistically significant differences in the number of women and men between the study and control groups.

The Discussion chapter is short a and it is not appropriately discuss the results. It is just a brief review of the field without showing the significance of these results. 

Thank you for comment. Discussion section has been improved. Due to the multitude of corrections in discussion section, they were not marked in the text.

Reviewer 2 Report

This is a very interesting article with regard to the potential impact of using Kinesio Tape on non-infectious complications after impacted mandibular third molar surgery. Nevertheless, minor revisions could improve article's quality.

  1. In the Introduction section the number of paeagraphs should be reduced. Relevant information should be added in the Discussion section.
  2. Tables' quality is acceptable.
  3. Newly published manuscripts should be included in the Discussion section.
  4. Grammatical errors should be corrected throughout the Text.

Author Response

Dear Reviewer,

We would like to thank you for your valuable comments on the article. Below you will find our reply to your review. All changes are with a description or a comment and changes have been made to the manuscript. Changes are marked with yellow.

This is a very interesting article with regard to the potential impact of using Kinesio Tape on non-infectious complications after impacted mandibular third molar surgery. Nevertheless, minor revisions could improve article's quality.

Thank You for comment.

In the Introduction section the number of paragraphs should be reduced. Relevant information should be added in the Discussion section.

Thank you for comment. The introduction has been shortened and some information has been moved to the discussion section. Due to the multitude of corrections in discussion section, they were not marked in the text.

Tables' quality is acceptable.

Thank You for comment.

Newly published manuscripts should be included in the Discussion section.

Thank you for comment. Discussion section has been improved. Due to the multitude of corrections in discussion section, they were not marked in the text.

Grammatical errors should be corrected throughout the Text.

Thank you for comment. We read our manuscript once again and made necessary language and grammar corrections to the text.

Reviewer 3 Report

Page 2 Line 53: “They are much more common in the extraction of lower wisdom teeth and normally are dolore post-extraction pains […]. Remove the word “dolor” from the sentence.

Page 2 Line 54: up to 90%. Remove the typeface “o”

Page 2 Line 56: Insert the determinative article “the” before “post-extraction”

Page 2 Line 58: This period is too long and not clear. Please, rewrite it.

Page 3 Line 105: Insert the article “the” before “continuity”

Materials and Method

Patient Recruitment and Study Groups

It would be advisable to indicate if and how many patients were symptomatic before the surgical procedure: jaw opening, pain intensity, swelling could be present even before the tooth removal and conditioning the postoperative course. Moreover, it is not specified whether the patients recruited into the study had received therapy prior to surgery.

Surgery

Indicate the type of anesthesia performed (plexus/troncular).

Among the main factors influencing the degree of postoperative swelling there are certainly the degree of inclusion of the wisdom tooth, the amount of osteoplasty performed and, above all, the type and size of the mucoperiosteal flap prepared. Therefore, these parameters should be reported and this lack impairs the informativeness of this study, in which the surgical difficulty is linked to operation time only. If they were specified later in a qualitative but not quantitative way, some of the methodological accuracy of the study would be affected.

Have any haemostatic materials been used (eg, fibrin sponge)?

Kinesiotaping

In the present study, the degree of tension in the tapes was approximately 15% of the maximum stretch. How was this degree of tape tension measured?

Author Response

Dear Reviewer,

We would like to thank you for your valuable comments on the article. Below you will find our reply to your review. All changes are with a description or a comment and changes have been made to the manuscript. Changes are marked with yellow.

Page 2 Line 53: “They are much more common in the extraction of lower wisdom teeth and normally are dolore post-extraction pains […]. Remove the word “dolor” from the sentence. 

Page 2 Line 54: up to 90%. Remove the typeface “o” 

Page 2 Line 56: Insert the determinative article “the” before “post-extraction”

Page 2 Line 58: This period is too long and not clear. Please, rewrite it. 

Page 3 Line 105: Insert the article “the” before “continuity”

Thank You for the remarks. I introduced your suggestion to the text. 

Materials and Method

Patient Recruitment and Study Groups

It would be advisable to indicate if and how many patients were symptomatic before the surgical procedure: jaw opening, pain intensity, swelling could be present even before the tooth removal and conditioning the postoperative course. Moreover, it is not specified whether the patients recruited into the study had received therapy prior to surgery.

All patients were asymptomatic and received no preoperative therapy. Only ketoprofen was given in a 100 mg dose taken twice daily.

Surgery

Indicate the type of anesthesia performed (plexus/troncular).

The surgical procedure was performed under the local anesthesia of Inferior Alveolar Nerve and buccal nerve using 2% lidocaine with 1:80000 noradrenaline.

Among the main factors influencing the degree of postoperative swelling there are certainly the degree of inclusion of the wisdom tooth, the amount of osteoplasty performed and, above all, the type and size of the mucoperiosteal flap prepared. Therefore, these parameters should be reported and this lack impairs the informativeness of this study, in which the surgical difficulty is linked to operation time only. If they were specified later in a qualitative but not quantitative way, some of the methodological accuracy of the study would be affected.

The incision was made on the ridge of the mandibular alveolus lateral from the linea obliqua towards the second molar. Then a release incision was made in the distal part of second molar, downward and in mesial direction for approximately 1 cm according to the same repeatable scheme for each patient. Each surgical extraction of the retained third molar in the mandible required the removal of the bone cover. 

Have any haemostatic materials been used (eg, fibrin sponge)?

After the surgery, the wound was closed with sutures for a period of 7 days. No additive haemostatic materials were used.

Kinesiotaping

In the present study, the degree of tension in the tapes was approximately 15% of the maximum stretch. How was this degree of tape tension measured?

Tape length (x) was calculated by equation (1). The distance from the area of supraclavicular lymph nodes to line A on the patient's face (y) was 115%. The tape length needed to create 15% tension was calculated from the formula:

 no. 1 in text (1).

The tape was prepared according to the calculations, which was about 86.96% of the y-distance. During application, the tape was stretched in such a way that it spanned the entire y-distance.

Round 2

Reviewer 1 Report

After the improvement and the corrections of the manuscript it reaches the standars of IJERPH and can be accepted in this form.

Author Response

Thank you very much for your valuable comments, which allowed us to improve our manuscript.  

Reviewer 3 Report

The authors followed the recommendations and the paper improved significantly. However, I ask to specify the absence of haemostatic material in paragraph 2.2 (surgery). Once the correction is made, I believe that the paper is suitable for publication in IJERPH.

Author Response

Thank you very much for your valuable comments, which allowed us to improve our manuscript. We added this information to the manuscript.  Participation in the research project was offered to 100 consecutive patients undergoing the surgical removal of the asymptomatic, impacted mandibular third molar, usually for orthodontic reasons. The exclusion criteria included unstable arterial hypertension. All patients were healthy, without contraindications for outpatient surgery. The risk of increased bleeding was not expected in patients. No additional hemostatic materials were used to exclude and minimize the influence of other factors on the test results. Therefore, the next step in our research could be the comparison of the use and no use of additional local hemostatic agents.